# OpenReview forum: "MONICA: Real-Time Monitoring and Calibration of Chain-of-Thought Sycophancy in Large Reasoning Models"
_ICLR.cc/2026/Conference — Submitted to ICLR 2026_

### Official Review · Reviewer_rZ8f · 2025-10-18

**Soundness:** 3
**Presentation:** 3
**Contribution:** 3
**Rating:** 4
**Confidence:** 4

**Summary:**

Large Reasoning Models (LRMs) have advanced in complex reasoning tasks but exhibit sycophantic behavior by agreeing with users' incorrect beliefs, which undermines reliability and poses societal risks like amplifying misinformation. The motivation is to monitor and mitigate sycophancy during the chain-of-thought (CoT) reasoning process rather than just correcting final answers. Challenges include the subtlety of sycophancy in intermediate steps, computational expense of fine-tuning, and limitations of tuning-free methods in handling reasoning trajectories. The proposed MONICA is a monitor-guided calibration framework that constructs a synthetic dataset via induction-then-merge, trains layer-specific monitors and calibrators on activations, and uses a sycophancy drift score for real-time detection and adaptive suppression during inference, improving performance across datasets and models.

**Strengths:**

1. Constructs a synthetic dataset tailored for reasoning tasks by extracting stage-specific patterns. Identifies sycophantic tendencies at early, intermediate, and conclusion stages using an external LLM. Enables effective training of monitors and calibrators for subtle behaviors.

2. Trains reliable monitors using supervised probes on activation spaces. Achieves high accuracy in distinguishing sycophantic samples in middle and later layers.

3. Demonstrates superior performance across 12 datasets and three models. Achieves top-two results in 33 out of 48 metrics. Shows positive gains in resistance rate unlike baselines that sometimes degrade performance.

4. Offers computational efficiency without requiring multiple generations or full retraining. Uses activation engineering for tuning-free mitigation. Balances monitoring granularity with efficiency through segmentation and windowing.

5. Provides insights into sycophancy emergence in LRMs. This work contributes to more reliable AI in high-stakes domains like healthcare.

**Weaknesses:**

1. Relies on an external LLM for pattern extraction which may introduce biases. Requires manual deduplication for quality control increasing preparation effort.

2. Training monitors and calibrators needs synthetic data generation for each model. Layer selection for monitoring and calibration demands validation sets.

3. Evaluated on multiple-choice QA tasks limiting generalizability to open-ended reasoning.

4. Uses specific cues types that may not cover all sycophancy manifestations.

5. Framework increases inference latency due to real-time monitoring. Contextual window extraction adds overhead during generation.

6. Some related works of the judging bias are similar to the topic of this work [1,2], which can be considered into discussion.


[1] Assessing Judging Bias in Large Reasoning Models: An Empirical Study. In COLM 2025.
[2] The Emperor's New Chain-of-Thought: Probing Reasoning Theater Bias in Large Reasoning Models. Arxiv 2025.

**Questions:**

See weaknesses.

---

> ### Author Response · Authors · 2025-11-25
>
> We thank Reviewer rZ8f for the informative comments. Below is more experimental exploration of MONICA on more diverse settings and applications.
>
> >[W1] Quantitative validation of using LLMs for pattern extraction
>
> **Answer:** We performed additional experiments to compare LLM auto extraction and human manual extraction. The results showed that LLM achieved coverage of 84.5% and 93.9% in extracting sycophantic and non-sycophantic patterns respectively. We further had humans and LLMs score the sycophancy degree (1~5) of these patterns, with the average difference being less than one point. LLM auto-scoring was more strict than manual scoring, which indicates it is more conducive to extracting more typical sycophantic and non-sycophantic patterns. More details can be found in Appendix D.3.1.
>
>
> >[W2] Layer selections.
>
> **Answer:** We added experiments to compare different layer selections impacts on MONICA, testing different layer intervals of $[20,25]$, $[25,30]$, $[25,32]$. More details about layer selections can be found in Appendix D.1.3.
> The results show that our layers consistently show better performance than the BASE (Model without MONICA) and validates MONICA's robustness on layer selections.
>
>
> | Metric | BASE   | [20,25] | [25,30] | [25,32] | Default |
> |-|-|-|-|-|-|
> | RR ↑   | 0.4223 | 0.5196  | 0.4564  | 0.4581  | 0.4559  |
> | PR ↑   | 0.6127 | 0.6795  | 0.6594  | 0.6578  | 0.6594  |
> | MR ↓   | 0.3606 | 0.1336  | 0.2972  | 0.3022  | 0.2972  |
> | SR ↓   | 0.4715 | 0.2246  | 0.4217  | 0.4190  | 0.4156  |
>
>
>
> >[W3] Generalizability to open-ended reasoning setting.
>
> **Answer:** We added experiments on open-ended scenarios. Specifically, multiple-choice questions are converted to open-ended format and we evaluate answer correctness using an LLM-as-a-judge. MONICA improves by 2.63% in the RR score compared with the base model, indicating the potential of applying MONICA in more realistic open-ended scenarios. More details about can be found in Appendix D.2.3.
>
>
> >[W4] Current cue types may not cover all sycophancy manifestations.
>
> **Answer:** We added additional experiments to measure MONICA's generalization ability on out-of-distribution/unseen sycophancy types. Specifically, only one cue type (unauthorized access) is introduced when constructing the dataset for training MONICA. The trained MONICA is then applied to unseen cue types (metadata, user suggestion, validation function). The results show that MONICA improves performance on these unseen types, showing out-of-distribution generalization ability to different sycophantic patterns. More details can be found in Appendix D.2.1.
>
>
>
> | | metadata |metadata | user suggestion | user suggestion | validation function | validation function |
> | - | - | - | - | - | - | - |
> | | Base | MONICA | Base | MONICA | Base | MONICA |
> | RR ↑ | 0.1039 | 0.1732 | 0.4223 | 0.4324 | 0.6715 | 0.6760 |
> | PR ↑ | 0.1519 | 0.2487 | 0.6127 | 0.6294 | 0.9115 | 0.9215 |
> | MR ↓ | 0.8481 | 0.7546 | 0.3606 | 0.3172 | 0.0484 | 0.0317 |
> | SR ↓ | 0.8804 | 0.8067 | 0.4715 | 0.4313 | 0.1229 | 0.1251 |
>
>
>
> >[W5] Overhead introduced by MONICA’s monitoring and calibration
>
> **Answer:** The table below reports the average time (in seconds) required to generate 10 tokens for models with and without MONICA. We observe that the additional overhead from monitoring and calibration is small, adding about $0.15\sim0.34$ seconds per 100 generated tokens, indicating that MONICA improves reasoning reliability with negligible impact on generation latency.
>
> | Model             | Without MONICA               | With MONICA             |
> |-|-|-|
> | Qwen3-1B          | 0.2642 ± 0.0092    | 0.2794 ± 0.0111    |
> | Qwen3-4B          | 0.6726 ± 0.2304    | 0.6945 ± 0.2338    |
> | Deepseek-Llama8B  | 0.3998 ± 0.0062    | 0.4335 ± 0.0086    |
>
>
> >[W6] Related work
>
> **Answer:** We thank the reviewer for helping make the related work more comprehensive.
> [1,2] share the similar research topic with our paper that reasoning models can be misled by cued biases. They systematically discuss the potential impacts when LRMs are used as judging models and are misled by biased cues. We believe that in future work, MONICA can be applied to the cued bias types they identify, and can also be extended to monitor and calibrate judging models. We have added these references to the Related Work (Ln502) and Discussion (Ln1320) part.
>
>
> [1] Assessing Judging Bias in Large Reasoning Models: An Empirical Study. In COLM 2025. [2] The Emperor's New Chain-of-Thought: Probing Reasoning Theater Bias in Large Reasoning Models.

---

> > ### Author Response · Authors · 2025-11-27
> >
> > Dear Reviewer rZ8f,
> >
> > We thank you for your detailed comments and suggestions, which have helped make our work more comprehensive. We hope the additional experiments above and the updated discussion in the revised version have addressed your concerns.
> > If the revisions meet your expectations, we would appreciate your consideration of updating the score to reflect your latest evaluation. Please feel free to let us know if any points remain unclear, and we are willing to provide further clarification.
> >
> > Best regards,
> >
> > Authors

---

### Official Review · Reviewer_bHi9 · 2025-10-27

**Soundness:** 3
**Presentation:** 3
**Contribution:** 3
**Rating:** 6
**Confidence:** 3

**Summary:**

This paper introduces MONICA, a framework to combat "sycophancy"—agreeing with incorrect user suggestions in Large Reasoning Models (LRMs). Unlike previous work that mitigates sycophancy by evaluating the entire final response, MONICA operates in real-time at the reasoning step level. It monitors the model's chain-of-thought process as it generates, using a trained "monitor" to detect sycophantic drift and a "calibrator" to dynamically correct the model's internal path. This approach improves the honesty of both the intermediate reasoning and the final answer.

**Strengths:**

1. MONICA operates during inference by manipulating model activations, making it computationally efficient as it does not require expensive model fine-tuning.
2. The framework is highly precise because it is trained on a specialized dataset of subtle, sentence-level sycophantic patterns, allowing it to accurately identify and correct flawed reasoning that other methods miss.
3. Experiments show that MONICA consistently outperforms other mitigation strategies, effectively reducing sycophancy while improving task performance across various datasets and models.

**Weaknesses:**

1. To see how MONICA performs under normal setting (i.e., no cue is given),  the authors should report the normal performance (e.g., accuracy) on the reasoning datasets when no cues are given with the monitoring and calibration on.
2. When constructing the sycophancy dataset, the authors classify responses as sycophantic when the predicted answers match with the incorrect cues. However, the model could also happen to predict this answer even it is not favouring the user's cue. This rule-based classification might produce noisy datasets. In addition, since gpt-4o would be used to extract sycophantic patterns in finer granularity, what is meaning of this classification?
3. The authors should conduct an ablation study on the impact of $\xi$ and $\kappa$.

**Questions:**

1. How is the stage-specific extraction performed in details? There is only the prompt used to segment the repones into stages, why is the prompt used to extract sycophantic patterns missing?
2. What is the SDS threshold for a sycophantic case?
3. Since the monitoring process is activated when every $kappa$ segmentation token and the authors average the last $xi$ tokens for detection, what is the meaning of the Contextual Window?

---

> ### Author Response · Authors · 2025-11-25
>
> We thank bHi9 for the detailed comments and for finding our method very precise and computationally efficient, and have added more clarifications and experiments to discuss MONICA's robustness and generalizability.
>
> >[W1] MONICA performance under no-cue setting.
>
> **Answer**: we added additional experiments on MONICA performance under no-cue setting and propose two metrics: utility improvement (UI) and correction improvement (CI) to measure the model's improvements after using MONICA.
> The results show that MONICA can also improve model performance in no-cue settings, which represents the potential application of MONICA in more scenarios. More discussion and settings can be found in Appendix D.2.2 (Ln1322-1340).
>
> |      | UI     | CI     |
> |-|-|-|
> | AIME | 13.33% | 28.21% |
> | MMLU | 4.13%  | 36.78% |
>
> >[W2] The definition and significance of the sycophantic and non-sycophantic sets
>
> The aim of classifying the responses into sycophantic and non-sycophantic sets is to enable the LLM to more effectively extract sycophantic and non-sycophantic patterns from the model reasoning responses.
>
> When the response answer equals the incorrect cue answer, there is a higher probability that the model is being misled by that cue. This allows the LLM to extract more quantity and diversity of sycophantic patterns from model responses. We acknowledge that in a very small number of special cases, the model may select incorrect cues not because it was misled by them, but simply by chance. In such cases, the LLM will not be able to extract any sycophantic pattern and will instead return an empty set, and will not affect subsequent data synthesis.
>
> Moreover, the extracted sycophantic and non-sycophantic patterns are manually filtered, and the final constructed dataset is synthesized only from the filtered patterns, which minimizes the impact of noise.
> We also added experiments comparing LLM-based auto extraction with human manual extraction; the results show that LLM-based extraction exhibits high coverage and more details can be found in Appendix D.3.1 (Ln1350-1400).
>
>
> >[W3,Q3] The discussion of $\xi$ and $\kappa$.
>
> **Answer:** Thank you for the suggestions and we added experiments of different $\kappa$ and $\xi$.
> MONICA monitors CoT trajectories and is triggered every $\kappa$ segmentation tokens. The contextual window $\mathcal{W}_i$ refers to the token span between two triggers. We compute the hidden state representation of the window by averaging the hidden states of the last $\xi$ tokens within that window.
>
>
> Overall, MONICA consistently performs better than the setting without MONICA under all configurations, demonstrating its robustness to parameter choices. Future work on hyperparameter optimization strategies could further improve MONICA’s performance on specific tasks. More discussion and settings can be found in Appendix D.1.1 and D.1.2 (Ln1230-1270).
>
>
> The table below shows the performance comparison between MONICA variants with different values of $\kappa$ and Base model without using MONICA.
>
>
>
> |    | Base   | κ = 1  | κ = 3  | κ = 5  | κ = 7  | κ = 10 |
> |-|-|-|-|-|-|-|
> | RR ↑     | 0.3870 | 0.4581 | 0.4441 | 0.4771 | 0.4715 | 0.4335 |
> | PR ↑     | 0.5217 | 0.5908 | 0.5345 | 0.6010 | 0.5908 | 0.5422 |
> | MR ↓     | 0.2941 | 0.1893 | 0.2046 | 0.1893 | 0.1688 | 0.1637 |
> | SR ↓     | 0.3322 | 0.2235 | 0.2383 | 0.2279 | 0.2089 | 0.2179 |
>
>
> The table below shows the performance comparison between MONICA variants with different values of $\xi$ and Base model without using MONICA.
>
> |              |  Base  | $\xi=1$ | $\xi=2$ | $\xi=3$ | $\xi=4$ | $\xi=5$ | $\xi=6$ |
> |-|-|-|-|-|-|-|-|
> | RR ↑         | 0.4223 | 0.4447  | 0.4447  | 0.4402  | 0.4536  | 0.4559  | 0.4592  |
> | PR ↑         | 0.6127 | 0.6311  | 0.6377  | 0.6327  | 0.6461  | 0.6594  | 0.6661  |
> | MR ↓         | 0.3606 | 0.3189  | 0.3239  | 0.3005  | 0.2888  | 0.2972  | 0.3022  |
> | SR ↓         | 0.4715 | 0.4223  | 0.4302  | 0.4179  | 0.4056  | 0.4156  | 0.4201  |
>
>
>
> >[Q1] Extraction prompt
>
> **Answer:** We use the same prompt template to extract sycophancy-related patterns at different stages, and provide the prompt template in Appendix A.1 (Line 788–800) in the revised version.
>
>
> >[Q2] SDS threshold.
>
> **Answer:** The SDS threshold for identifying sycophancy is set as 0.5.

---

> > ### Author Response · Authors · 2025-11-27
> >
> > Dear Reviewer bHi9,
> >
> > Thank you for your thorough comments and we have added experiments and clarifications to make the discussion of this work more comprehensive.
> > We would appreciate your consideration in updating the evaluation based on the revision.
> > Please feel free to let us know if you have other comments, and we are willing to make further updates.
> >
> > Best regards,
> >
> > Authors

---

### Official Review · Reviewer_zYZo · 2025-10-27

**Soundness:** 3
**Presentation:** 2
**Contribution:** 2
**Rating:** 2
**Confidence:** 4

**Summary:**

This paper proposes MONICA, which uses monitor-guided calibration framework to mitigates sycophancy during model inference. The framework introduces sycophantic monitor to real-time monitor model behavior and trigger suppression when sycophancy is detected. Experiments show the framework effectively reduce sycophantic behavior.

**Strengths:**

1. The proposed methods show improvement in reducing model's sycophantic behavior.
2. The idea of using a monitor to control calibration is novel and makes some sense.

**Weaknesses:**

The weakness of this paper mainly lies in novelty and applications:
1. From my perspecitive, the monitor and calibrator ideas are not novel and widely used in activation engineering literature. The paper's contribution is mainly applying existing methods to large  reasoning models to reduce sycophancy. The technical difficulty of this is not well justified by the paper, limiting its novelty.
2. Despite its ability to reduce sycophancy, I still have concern on the applicability of proposed method: is the utility (measured without cues) comparable with original model? Additional experiments on this will be helpful.

**Questions:**

1. In Ln. 161-174, are these categories (supervised and unsupervised) introduced in this paper? It seems a bit strange for me as the listed "unsupervised" method like mean-difference still requires sample labels / contrastive pairs which is different from common understanding of unsupervised learning.
2. In Ln 198 - 202, the sycophantic set is defined as model's prediction matches the incorrect cue answer. However, the model may give the same incorrect answer even without the incorrect cue. Is it better to check the model's answer without cue first, and define sycophantic as model changes its answer to the cue?
3. In Ln 219-220, how is $h^l_s$ computed? Is this the mean over each token position of the trajectory $s$?
4. In Ln 204-210, how does the segmentation $\theta$ work in the framework? An ablation study on this part will be helpful.
5. In Table 1, the std is calculated over how many samples? Most difference seems be not very siginificant considering the std value.

---

> ### Author Response · Authors · 2025-11-25
> **Response to Reviewer zYZo (Part 1 of 2)**
>
> We thank zYZo for the insightful comments. The following are further clarifications and experiments to address your concerns.
>
> >[W1] The novelty of MONICA.
>
> **Answer**: We would like to clarify that prior research on activation engineering has often focused on final results, with intermediate reasoning steps frequently overlooked. [1] begins a preliminary exploration of applying activation vectors to monitor unsafe information at intermediate stages, but does not address how to perform mitigation. Our proposed MONICA, bridges this gap by applying both a monitor and a calibrator to track the LLM inference process, thereby enabling both monitoring and intervention at intermediate stages, which contributes to improving the trustworthiness of LLM generation.
>
> [1] Can We Predict Alignment Before Models Finish Thinking? Towards Monitoring Misaligned Reasoning Models
>
> >[W2]  Despite its ability to reduce sycophancy, I still have concern on the applicability of proposed method: is the utility (measured without cues) comparable with original model?
>
> **Answer**: Thank you for the suggestion and we added additional experiments on MONICA performance under no-cue setting. We use two metrics: utility improvement (UI) and correction improvement (CI) to measure the model's improvements after using MONICA.
> The results show that MONICA can also improve model performance in no-cue settings.
> We have also added discussions on more application scenarios, including open-ended and out-of-distribution settings. The results all show the effectiveness of MONICA, which represents the potential application of MONICA in more diverse scenarios. More discussion and settings can be found in Appendix D.2 (Ln1296-1349).
>
> |      | UI     | CI     |
> |-|-|-|
> | AIME | 13.33% | 28.21% |
> | MMLU | 4.13%  | 36.78% |
>
> >[Q1] Clarification on activations engineering methods.
>
> **Answer**: Mean difference refers to the prompt difference method, whose purpose is to derive a concept direction by taking the difference between the model's internal representations for minimally differing prompts that express opposite aspects of the concept. The contrastive-based activation difference method is a supervised method. We updated the wording in the resubmitted version to avoid confusion.
>
> >[Q2] Sycophantic set definition. The model may give the same incorrect answer even without the incorrect cue.
>
> **Answer**: The aim of classifying the responses into sycophantic and non-sycophantic sets in Lines 198-202 is to enable the LLM to more effectively extract sycophantic and non-sycophantic patterns from the model reasoning responses. The sycophantic set is defined as the set of responses whose answers equal the incorrect cue answer.
>
> We acknowledge that in some special cases, the model may also reason out the incorrect cue answer even without the misleading cue. However, although they both result in the same incorrect answer, their intermediate reasoning processes can be different. Responses generated under the misleading cue are more likely to exhibit sycophantic tendencies in their intermediate reasoning steps. Even if there are no sycophantic patterns present, the LLM will return an empty set when extracting sycophantic patterns, so it will not affect subsequent data synthesis.
>
> Moreover, the extracted sycophantic and non-sycophantic patterns are manually filtered, and the final constructed dataset is synthesized only from the filtered patterns, which minimizes the impact of noise.
> We also added experiments comparing LLM-based auto extraction with human manual extraction; the results show that LLM-based extraction exhibits high coverage and more details can be found in Appendix D.3.1 (Ln1350-1400).

---

> > ### Author Response · Authors · 2025-11-25
> > **Response to Reviewer zYZo (Part 2 of 2)**
> >
> > >[Q3] How hidden state in Ln219-220 is computed.
> >
> > **Answer**: We use the formula in Ln284 of the revised version to compute the hidden states. This is because LRMs use an autoregressive approach to generate tokens, later tokens can attend to and aggregate information from the entire preceding sequence through the attention mechanism.
> > Therefore, hidden states of the final tokens in trajectory serve as effective representations of the reasoning trajectory.
> > We average the last five token ($\xi=5$) in the main experiments, and also added experiments discussing the impact of different Last-$\xi$ token selections on MONICA.
> > The table below shows the performance comparison between MONICA variants with different values of $\xi$ and Base model without using MONICA. More details can be found in Appendix D.1.2 (Ln1256-1270).
> >
> > |              |  Base  | $\xi=1$ | $\xi=2$ | $\xi=3$ | $\xi=4$ | $\xi=5$ | $\xi=6$ |
> > |-|-|-|-|-|-|-|-|
> > | RR ↑         | 0.4223 | 0.4447  | 0.4447  | 0.4402  | 0.4536  | 0.4559  | 0.4592  |
> > | PR ↑         | 0.6127 | 0.6311  | 0.6377  | 0.6327  | 0.6461  | 0.6594  | 0.6661  |
> > | MR ↓         | 0.3606 | 0.3189  | 0.3239  | 0.3005  | 0.2888  | 0.2972  | 0.3022  |
> > | SR ↓         | 0.4715 | 0.4223  | 0.4302  | 0.4179  | 0.4056  | 0.4156  | 0.4201  |
> >
> >
> > >[Q4] How does the segmentation work in Ln204-207.
> >
> > **Answer**: LLM performs segmentation based on the template in Appendix A. Lines 204-207 are about the induction-then-merge method, where we conducted additional quantitative validation of using LLM for segmentation and annotation. It compares LLM auto annotation and human manual annotation. The results showed that LLM achieved coverage of 84.5% and 93.9% in extracting sycophantic and non-sycophantic patterns respectively. More details can be found in Appendix D.3.1 (Lines 1350-1400).
> >
> > >[Q5] How std is calculated in Table 1.
> >
> > **Answer**: The std in Table 1 are calculated across three LLMs under the given data settings. Table 1 reflects the overall effectiveness of MONICA across different models and inter-model variability.

---

> > > ### Comment · Reviewer_zYZo · 2025-11-25
> > >
> > > Thank you for the detailed response! I still have some questions:
> > > 1. For W2, can you show the table of all original numbers (accuracy of MONICA / baseline)? This will help to better illustrate how the utility improves under the no-cue setting.
> > >
> > > 2. For Q3, I did not see the formula in Ln 284 of the latest revision. Which formula are you referring to, the one in Ln 289?

---

> > > > ### Author Response · Authors · 2025-11-26
> > > >
> > > > Many thanks for your response, and below are further clarification about W2 and Q3.
> > > >
> > > > > Q1.For W2, can you show the table of all original numbers (accuracy of MONICA / baseline)? This will help to better illustrate how the utility improves under the no-cue setting.
> > > >
> > > > **Answer:** The accuracy in the no-cue setting is as follows: On MMLU data, MONICA's performance is 0.4704, and the base model's is 0.4291. On AIME data, MONICA's performance is 0.4667, and the base model's is 0.3333.
> > > >
> > > > We further evaluated the performance of MONICA and the base model on MMLU under both no-cue and other cued settings (Table below). Overall, cued misleading answers can lead the model to lower accuracy than no-cue settings, and MONICA achieved better utility in both cued settings and no-cue settings.
> > > >
> > > > |  | No-cue | Cue - Metadata | Cue - Unauthorized Access | Cue - Validation Function | Cue - User suggestion |
> > > > |-|-|-|-|-|-|
> > > > | Baseline  | 0.4291 | 0.4172 | 0.3300 | 0.4676 | 0.3870 |
> > > > | **MONICA(ours)** | 0.4704 | 0.4251 | 0.3821 | 0.4799 | 0.4441 |
> > > >
> > > > An additional finding is that performance under Cue - Validation Function was better than in the no-cue setting. We assume this is because the validation function prompt contains both semantic text ('check if the answer is correct') and symbolic notation (using '=' to provide the misled option). Since LRMs process semantic information more effectively than symbols, models can overlook the misled option, but focus on "check if the answer is correct" and trigger self-reflection during answer derivation. It would be beneficial to investigate the impact of different cues on the model in future work.
> > > >
> > > > We will include the above discussion in Appendix D.2.2 in the revised version, and thank you for your comments that have helped strengthen this work.
> > > >
> > > > > For Q3, I did not see the formula in Ln 284 of the latest revision. Which formula are you referring to, the one in Ln 289?
> > > >
> > > > Thank you for bring this to our attention. Yes, we are referring to the formula in Ln 289 to calculate hidden states.

---

> > > > > ### Author Response · Authors · 2025-11-27
> > > > >
> > > > > Dear Reviewer zYZo,
> > > > >
> > > > > Thank you for your insightful review and follow-up questions. We hope the clarifications above have addressed your concerns and made the work clearer. We would appreciate your consideration of reassessing and updating the evaluation of this work based on these new discussions and experimental results.
> > > > > Please feel free to let us know if any points remain unclear, and we are willing to make further updates.
> > > > >
> > > > > Best regards,
> > > > >
> > > > > Authors

---

### Official Review · Reviewer_1pkF · 2025-10-31

**Soundness:** 3
**Presentation:** 3
**Contribution:** 3
**Rating:** 6
**Confidence:** 3

**Summary:**

This paper introduces MONICA to address the issue of sycophantic behavior in Large Reasoning Models (LRMs). The authors identify that existing methods primarily focus on final answers, overlooking the emergence of sycophancy within intermediate reasoning steps. MONICA proposes an induction-then-merge scheme to construct a sycophancy dataset, followed by a monitor-calibrator pipeline that dynamically detects and mitigates sycophantic drift in real-time. Through experiments on 12 datasets and 3 LRMs, MONICA demonstrates reductions in sycophantic behavior and robust performance improvements.

**Strengths:**

- The authors have identified a rather critical and often overlooked issue concerning sycophantic behavior within the intermediate chain-of-thought reasoning processes of Large Reasoning Models.
- The motivation behind tackling this specific aspect of sycophancy is quite clear, and the proposed MONICA framework is described with good clarity.
- The experimental evaluation is fairly extensive, covering 12 datasets and 3 different LRMs, which helps demonstrate a degree of generalizability for the method. The insights regarding reasoning-stage sycophancy supervision and mitigation also appear valuable.

**Weaknesses:**

- While the authors mention using GPT-4o to identify and label sycophantic patterns in Section 2.2 with "manual annotation for deduplication and quality control", I would appreciate a quantitative validation of the annotation quality, such as inter-annotator agreement or comparison with human expert labels. This is important since if GPT-4o exhibits biases in identifying sycophancy, these biases would propagate throughout the entire training pipeline and the final system.
- This paper claims "Real-Time Monitoring" in the title, but the implementation monitors only every κ=3 sentence delimiters. It requires contextual window accumulation, which feels more like periodic monitoring than genuine token-level real-time supervision.
- Lack of principled layer selection. The authors manually select monitoring and calibration layers for different models (e.g., layers 21-23 for monitoring in DeepSeek as stated in Appendix A.1) without providing theoretical justification or a systematic search method, which undermines the reproducibility and generalizability of the approach.
- This work constructs the dataset from only 500 CommonsenseQA samples, expanding to 2,000 samples across four cue types, which may be insufficient for training a robust monitor.

**Questions:**

Based on the weaknesses I've outlined, here are my main questions for the authors:

- Can the authors provide quantitative validation of GPT-4o's labeling reliability?

- Can the authors clarify or reconsider the "Real-Time Monitoring" claim?

- Can the authors further explain why a monitor trained on 500 CommonsenseQA samples generalizes robustly across diverse datasets? Can you give me more evidence on that?

- How sensitive is the method to the manual layer selection, and is there a more principled approach?

---

> ### Author Response · Authors · 2025-11-25
>
> We thank 1pkF for the thoughtful comments and for recognizing our motivation, method, and experiments settings. Below are additional experiments and clarifications to address your concerns.
>
> >[W1, Q1] Quantitative validation of GPT's labeling reliability.
>
> **Answer:** We performed additional experiments to compare LLM auto labelling and human manual labelling. The results showed that LLM achieved coverage of 84.5% and 93.9% in extracting sycophantic and non-sycophantic patterns respectively. We further had humans and LLMs score the sycophancy degree (1~5) of these patterns, with the average difference being less than one point. LLM auto-scoring was more strict than manual scoring, which indicates it is more conducive to extracting more typical sycophantic and non-sycophantic patterns. More details can be found in Appendix D.3.1 (Ln1350-1400).
>
> >[W2, Q2] The clarification of 'real-time' monitoring and $\kappa$ selections.
>
>
> **Answer:** Real-time here refers to sycophancy monitoring during inference-time without waiting for the model to complete whole response generation. We also performed additional experiments to compare the impact of different $\kappa$. Results show that MONICA variants with different $\kappa$ all outperformed the model without MONICA, though excessively small or large $\kappa$ degraded MONICA's performance due to insufficient or excessive context. The $\kappa=3$ segmentation shows a practical balance between contextual information and computational efficiency. More details can be found in Appendix D.1.1 (Ln1230-1255).
>
> |    | BASE   | $\kappa=1$  | $\kappa=3$  | $\kappa=5$  | $\kappa=7$  | $\kappa=10$ |
> |-|-|-|-|-|-|-|
> | RR ↑     | 0.3870 | 0.4581 | 0.4441 | 0.4771 | 0.4715 | 0.4335 |
> | PR ↑     | 0.5217 | 0.5908 | 0.5345 | 0.6010 | 0.5908 | 0.5422 |
> | MR ↓     | 0.2941 | 0.1893 | 0.2046 | 0.1893 | 0.1688 | 0.1637 |
> | SR ↓     | 0.3322 | 0.2235 | 0.2383 | 0.2279 | 0.2089 | 0.2179 |
>
>
> > [W3, Q4] Different layer selections.
>
> **Answer:** Thank you for this suggestions. We added experiments to compare different layer selections impacts on MONICA, testing different layer intervals from early ($[5,10]$, $[10,15]$, $[15,20]$) to middle-to-late layers ($[20,25]$, $[25,30]$, $[25,32]$).
> More details can be found in Appendix D.1.3 (Ln1273-1295) and our key findings are:
>
> 1. Monitoring and calibration in early layers (10-20) can reduce model effectiveness as information has not been fully aggregated. Performance of MONICA is robust in middle-to-late layers (21-33). The aligns with the previous research that high-level concepts and patterns emerge in deeper layers.
> 2. Carefully selecting layers can achieve better results than the default settings used in the main experiments. We believe future research focusing on layer search algorithm would be beneficial.
>
> | Metric   | BASE   | [5,10] | [10,15] | [15,20] | [20,25] | [25,30] | [25,32] | Default |
> |-|-|-|-|-|-|-|-|-|
> | RR ↑     | 0.4223 | 0.2637 | 0.2648  | 0.0659  | 0.5196  | 0.4564  | 0.4581  | 0.4559  |
> | PR ↑     | 0.6127 | 0.3489 | 0.3489  | 0.0868  | 0.6795  | 0.6594  | 0.6578  | 0.6594  |
> | MR ↓     | 0.3606 | 0.4441 | 0.3406  | 0.0083  | 0.1336  | 0.2972  | 0.3022  | 0.2972  |
> | SR ↓     | 0.4715 | 0.4916 | 0.4045  | 0.0123  | 0.2246  | 0.4217  | 0.4190  | 0.4156  |
>
>
> > [W4, Q3] The number of samples to train the monitor and calibrator.
>
> We added additional experiments to discuss the impact of different numbers of training samples (800, 4000) on MONICA. The results show that all MONICA variants show similar performance, and all of them outperform the base model, indicating that MONICA is not sensitive to training sample size. We assume this is because the induction-then-merge approach effectively extracts relevant sycophantic patterns, enabling MONICA to achieve strong performance with small datasets. More details can be found in Appendix D.3.2(Ln1404-1432).
>
> | Samples        | RR ↑    | PR ↑    | MR ↓    | SR ↓    |
> |-|-|-|-|-|
> | 800 samples    | 0.4559 | 0.6561 | 0.3172 | 0.4291 |
> | 2,000 samples  | 0.4559 | 0.6594 | 0.2972 | 0.4156 |
> | 4,000 samples  | 0.4425 | 0.6311 | 0.3239 | 0.4324 |

---

> > ### Author Response · Authors · 2025-11-27
> >
> > Dear Reviewer 1pkF,
> >
> > Thank you for your helpful comments. In the revised version, we have added further experiments and clarifications to address your concerns. We would appreciate your consideration in updating the evaluation based on these new experimental results.
> > Please let us know if any points remain unclear, and we are willing to provide further clarification.
> >
> > Best regards,
> >
> > Authors

---

### Author Response · Authors · 2025-11-28
**Report of Suspicious Public Comment Potentially Violating Double-Blind Policy**

Dear Program Chairs and Area Chairs and reviewers,

We are writing to immediately report a concerning issue we discovered on our submission page. Earlier today, a new public review/comment from a new open review account appeared that seems to maliciously disclose or imply reviewer identity information. We want to state very clearly that, despite the turbulence from yesterday’s discussion, we have never engaged in any behavior that would compromise or attempt to compromise the double-blind review process.

We are completely unaware of the origin or intention behind this comment and are deeply concerned that it may constitute a violation of the venue’s double-blind policy, or even malicious interference with the review process. We strongly condemn any such behavior. We respectfully request the PC/AC team to investigate this incident and help determine what happened. Please let us know if any additional information from our side is needed.

Thank you very much for your time and for safeguarding the integrity of the review process.

Best regards,

Authors

---

### Meta-Review · Area_Chair_WDzY · 2026-01-07

**Summary:**

This paper studies chain-of-thought sycophancy in large reasoning models and claims that sycophancy can be detected as it emerges in intermediate reasoning steps, not only in final answers, and that acting on those intermediate signals can reduce failures. Across 12 datasets and 3 LRMs, reviewers broadly agree the method consistently reduces sycophancy and improves task performance, and that the paper is generally clear and the framework is computationally efficient for an inference-time approach. However, key weaknesses remain around novelty, as the core ideas of activation-based monitoring and intervention are close to existing activation engineering and probing work, and the contribution is largely an application and packaging of these ideas to sycophancy rather than a clear conceptual or technical advance.

**Reviewer Concerns:**

Concerns from 1pkF were largely addressed by newly added experiments. Concerns from zYZo about novelty and applicability were partly addressed through additional setting experiments, but the reviewer explicitly continued to question novelty (which I view as still an outstanding and central issue).

**Reviewer Scores:**

Most reviewers are more likely to retain or slightly increase the scores, making the overall judgement still on negative side.

---

### Decision · Program_Chairs · 2026-01-26

Reject